# Monitoring Dark-State Dynamics of a Single Nitrogen-Vacancy Center in Nanodiamond by Auto-Correlation Spectroscopy: Photonionization and Recharging

**DOI:** 10.3390/nano11040979

**Published:** 2021-04-10

**Authors:** Mengdi Zhang, Bai-Yan Li, Jing Liu

**Affiliations:** 1Department of Pharmacology, College of Pharmacy, Harbin Medical University, Harbin 150081, China; mz10@iu.edu; 2Department of Physics, Indiana University-Purdue University Indianapolis, Indianapolis, IN 46202, USA

**Keywords:** NV centers, nanodiamond, blinking, ionization, auto-correlation

## Abstract

In this letter, the photon-induced charge conversion dynamics of a single Nitrogen-Vacancy (NV) center in nanodiamond between two charge states, negative (NV^−^) and neutral (NV^0^), is studied by the auto-correlation function. It is observed that the ionization of NV^−^ converts to NV^0^, which is regarded as the dark state of the NV^−^, leading to fluorescence intermittency in single NV centers. A new method, based on the auto-correlation calculation of the time-course fluorescence intensity from NV centers, was developed to quantify the transition kinetics and yielded the calculation of transition rates from NV^−^ to NV^0^ (ionization) and from NV^0^ to NV^−^ (recharging). Based on our experimental investigation, we found that the NV^−^-NV^0^ transition is wavelength-dependent, and more frequent transitions were observed when short-wavelength illumination was used. From the analysis of the auto-correlation curve, it is found that the transition time of NV^−^ to NV^0^ (ionization) is around 0.1 μs, but the transition time of NV^0^ to NV^−^ (recharging) is around 20 ms. Power-dependent measurements reveal that the ionization rate increases linearly with the laser power, while the recharging rate has a quadratic increase with the laser power. This difference suggests that the ionization in the NV center is a one-photon process, while the recharging of NV^0^ to NV^−^ is a two-photon process. This work, which offers theoretical and experimental explanations of the emission property of a single NV center, is expected to help the utilization of the NV center for quantum information science, quantum communication, and quantum bioimaging.

## 1. Introduction

A solid state non-classical light source, Nitrogen Vacancy (NV) centers in diamond have recently drawn unprecedented interest because of their unique spin states and remarkable single photon emission at room temperature [1,2]. NV centers have been utilized as sensors for nanoscale magnetic [3] and electric field sensing [4], fluorescence reporters for biomolecular labeling [5,6,7,8], modals in super resolution imaging [9,10,11,12], and the light source for quantum information processing. Compared with other single photon sources, such as quantum dots [13,14], the NV center is resistant to photo-bleaching. However, the intermittency of the fluorescence emission from a single NV center was also observed under several different conditions, leaving the NV center in a “bright state” or a “dark state”. 

This “dark state” is usually eluded in encoding the quantum bit, the accuracy of which is deteriorated by the fluorescence jump and spectral diffusion of single photon emitters [15]; in addition, the “dark state” also prevents efficient coupling between emitters and nano-cavities [16,17]. A precise control of the states of the luminescent NV center in diamond is crucial in the fields of quantum science [18,19]. However, just as a coin has two sides, the “dark state” is preferred in the super resolution imaging of nanodiamonds. The metastable “dark state” shelved the NV center from the excited state, leading to fluorescence intermittency in a time series, which allows for localizing individual NV centers in a single nanodiamond via a stochastic super-resolution scheme [9,10,11,12]. 

However, the mechanism of the fluorescence intermittency is still controversial, since many effects would contribute to the blinking behavior of the NV centers. The fluorescence intermittency of NV centers was first reported in a host of 5 nm nanodiamonds [20,21]; furthermore, researchers also found that the blinking behavior is size-dependent, and the fluorescence emission from NV centers changed from stable to intermittent when the size of the host-diamond was reduced from micrometers to nanometers by oxidation [22]. These experimental observations suggest that the oxidative groups or the impurities on the surface of the host-diamond would promote the tunneling of electrons of the NV center located within the diamond. Meanwhile, the wavelength-dependent blinking behavior of the NV center was reported by various groups and was also observed in our experiment (as shown below) [16,23,24]. This indicates that the charge conversion could happen between two charge states of the NV center [23]. As shown in Figure 1, the photon-induced ionization of the excited state of the negatively charged NV center (NV^−^) results in the suppression of the fluorescence emission, leading to the “dark state” [17,25], which corresponds to the neutrally charged NV center (NV^0^); meanwhile, the recharging of the NV^0^ can lead to the recovery of NV^−^, i.e., “bright state” [17,26]. Therefore, the NV center in the nanodiamond has switchable meta-states, both of which can emit fluorescent photons upon excitation [15,24]. Under proper excitation (532 nm), the transition between NV^−^ and NV^0^ happened as the fluorescence intermittency from a single NV center was observed (Figure 1c). Two distinct fluorescence states were observed (Figure 1c), and intensity-gated time-correlated single photon counting (TCSPC) curves are demonstrated in Figure 1d, showing a distinct fluorescence lifetime from the “ON” and “OFF” states. Therefore, tracking the dynamics of the “bright/dark” states of the NV center will contribute to the understanding of the photoemission mechanism of NV centers under different environmental conditions, from which various applications can benefit. 

The fluorescence intermittency is usually characterized by the ON/OFF histograms; however, the ON/OFF histogram highly relies on the threshold used to define the “ON” and “OFF” states, which are only distinguishable when the “ON” and “OFF” states are relatively long (milliseconds to seconds) [27]. It becomes difficult for the ON/OFF histogram to quantify the transition rate between bright and dark states when it occurs in the temporal range below milliseconds. In contrast, the auto-correlation spectroscopy (ACS) of fluorescence fluctuations has the flexibility to measure the dynamics in a broad temporal range (sub-microseconds to minutes) [28,29]. ACS was usually used to characterize the Brownian motion in solutions based on the signal fluctuation when the molecule moves in and out the focus volume [30,31,32]; in addition, it provides the dynamics of the triplet states when the pronation happens between the molecules and the solvent [29]. In this article, we utilized the ACS to reveal the blinking mechanism of the NV center in diamond.

In this work, we first deduce the analytical equation of the auto-correlation function of an immobilized emitter based on a three-level system. Optical properties of the single NV center were characterized in the time-resolved single molecule spectroscopy. It is demonstrated that the fluorescence intermittency in a single NV center is due to the transition between negative and neutral charged states; this transition is a two-photon procedure under a specific wavelength of excitation, as confirmed in the power-dependent measurements.

## 2. Theory

The typical electronic states of NV^−^ in the band gap of the diamond can be modeled as shown in Figure 2. It comprises three states, including the ground state |0>, excited state |1>, and dark state |2>. The transition rates between these states are defined as *k*_01_, the excitation rate from |0> to |1>; k_10_, the de-excitation (or spontaneous radiative emission) rate from |1> to |0>; k_12_, often defined as the inter-system transition rate from |1> to |2> via the non-radiative pathway; and k_20_, the relaxation rate from |2> to |0>. For a single NV center under proper excitation, the population of each state at time *t* can be obtained by solving the following differential equations.
(1)$$\frac{d}{dt}=\left\{\begin{array}{c} B_{0}\left(t\right)\\ B_{1}\left(t\right)\\ D_{2}\left(t\right) \end{array}\right\}=\left\{\begin{array}{ccc} -k_{01} & k_{10} & k_{20}\\ k_{01} & -k_{01}-k_{10} & 0\\ 0 & k_{12} & -k_{20} \end{array}\right\}\left\{\begin{array}{c} B_{0}\left(t\right)\\ B_{1}\left(t\right)\\ D_{2}\left(t\right) \end{array}\right\}$$
with *B*_0_(*t*), *B*_1_(*t*), and *D*_2_(*t*) representing the population at time *t* in state |0>, |1>, and |2>, respectively. They obey the condition B_0_(*t*) + B_1_(*t*) + D_2_(*t*) = 1, and have a boundary condition B_0_(0) = 1, B_1_(0) = D_2_(0) = 0. To simplify the above equation, we can take the assumption that k_10_ >> k_12_, k_20_, and in fact k_10_ is in the range of ns^−1^, while k_12_ and k_20_ are in the range of µs^−1^ [23]. By taking the assumption and the boundary conditions, one can solve Equation (1) to obtain the time-dependent population in each state in the µs^−1^~ms^−1^ range as follows.
(2)$$\begin{array}{l} B_{0}\left(t\right)=\frac{k_{10}k_{20}}{k_{01}\left(k_{12}+k_{20}\right)+k_{10}k_{20}}+\frac{k_{01}k_{10}k_{12}}{\left(k_{01}+k_{10}\right)\left[k_{01}\left(k_{12}+k_{20}\right)+k_{10}k_{20}\right]}\exp\left[-(k_{20}+\frac{k_{01}k_{12}}{k_{01}+k_{10}})\cdot t\right]\\ B_{1}\left(t\right)=\frac{k_{01}k_{20}}{k_{01}\left(k_{12}+k_{20}\right)+k_{10}k_{20}}+\frac{k_{01}^{2}k_{12}}{\left(k_{01}+k_{10}\right)\left[k_{01}\left(k_{12}+k_{20}\right)+k_{10}k_{20}\right]}\exp\left[-(k_{20}+\frac{k_{01}k_{12}}{k_{01}+k_{10}})\cdot t\right]\\ D_{2}\left(t\right)=\frac{k_{01}k_{12}}{k_{01}\left(k_{12}+k_{20}\right)+k_{10}k_{20}}-\frac{k_{01}k_{12}}{k_{01}\left(k_{12}+k_{20}\right)+k_{10}k_{20}}\cdot \exp\left[-(k_{20}+\frac{k_{01}k_{12}}{k_{01}+k_{10}})\cdot t\right] \end{array}$$

The constant terms in Equation (2) are regarded as the steady-state population of each population when time t→∞, and we define them as $B_{0}^{eq}$, $B_{1}^{eq}$ and $D_{2}^{eq}$, respectively. In addition, we also define $k_{20}+\frac{k_{01}k_{12}}{k_{01}+k_{10}}=1/\tau_{D}$, with $\tau_{D}$ as the lifetime of the dark state. The fluorescence intensity from a single NV center can be included as:(3)$$I\left(t\right)=k_{10}\eta B_{1}\left(t\right)=k_{10}\eta B_{1}^{eq}[\left(\frac{D_{2}^{eq}}{1-D_{2}^{eq}}\right)\exp\left(-\frac{t}{\tau_{D}}\right)+1]$$
Here η includes the contribution of quantum yield of the NV center, the collection efficiency of the system, and the quantum yield of the detector, and it is different from the case in which the molecule is diffusing in a solution where one needs to take the spatial distribution of the intensity into account. In our case, the molecule we are interested in is immobilized on a substrate. Thus, the fluorescence intensity fluctuations from single NV center will be
(4)$$\delta I\left(t\right)=I\left(t\right)-<I\left(t\right)>=k_{10}\eta \delta B_{1}\left(t\right)$$

And the auto-correlation function G(*τ*) will be
(5)$$G\left(\tau \right)=\frac{<\delta I\left(t\right)\delta I\left(t+\tau \right)>}{{<I\left(t\right)>}^{2}}=\frac{D_{2}^{eq}}{1-D_{2}^{eq}}\exp\left(-t/\tau_{D}\right)$$
Here the autocorrelation is calculated directly based on the intensity fluctuation, which is the result of multi-state transitions. Instead, one can derive the autocorrelation with ordered operators, which reflects the inter-state transitions [33], but a careful examination and validations are needed. Unlike the regular auto-correlation function of small molecules diffusing within a solution, the auto-correlation function here focuses on the fluorescence intensity fluctuation of an immobilized NV center; a spatial distribution of the excitation intensity will not influence the auto-correlation results, and therefore one can use any proper objective for excitation. While in regular fluorescent correlation spectroscopy (FCS), a small excitation volume from a high numerical-aperture objective is required. By fitting the auto-correlation curve with Equation (5), one can obtain parameters $D_{2}^{eq}$ and $\tau_{D}$, from which the rates k_12_ and k_20_ can be expressed as:(6)$$\begin{array}{ll} k_{12}=\frac{1}{\tau_{D}}\cdot \frac{k_{01}+k_{10}}{k_{10}}\cdot D_{2}^{eq} & \left(a\right)\\ k_{20}=\frac{1}{\tau_{D}}\cdot (1-D_{2}^{eq}) & \left(b\right) \end{array}$$

In Equation (6a), k_10_ can be approximated as $k_{10}=1/\tau_{B}$, and $\tau_{B}$ is the fluorescence lifetime obtained in time-correlated single photon counting (TCSPC) measurements. k_01_ is the excitation rate expressed as $k_{01}=\sigma \cdot I_{exc}/\hbar \nu$, with σ as the absorption cross-section, $I_{exc}$ as the excitation laser intensity, and ℏν as the photon energy of the excitation beam.

## 3. Results and Discussion

### 3.1. Selective Charge Ionization of Single NV Centers upon Excitation

The photon emission of a single NV center on the MgO substrate was systematically characterized under the excitation by the laser with different wavelengths (532 nm and 633 nm) (Figure 3). First of all, the single NV center exhibited differential emission profiles when being excited by the 532 nm and 633 nm lasers. As indicated in the time-course emission intensity trajectory (Figure 3a), the NV center had a two-states intermittency (Figure 3a, inset) when being excited by the 532 nm laser. For the same NV center, the 633 nm excitation shifted its emission profile to a mono-state. On the other hand, as expected, the anti-bunching effect was observed in the NV center for both the 532 nm and 633 nm excitations (Figure 3b). Meanwhile, the spontaneous decay curves suggest that the NV center has a similar excited-state lifetime under the 532 nm and 633 nm excitation (Figure 3c). All these evidences support our hypothesis that photonionization happens in the NV^−^ under a 532 nm excitation and leads to a “dark state” (NV^0^), and this process (ionization) is stopped when the excitation photon energy (for example, a 633 nm excitation) is smaller than the gap between the excited state and the conduction band [23]. Both “bright” (NV^−^) and “dark” (NV^0^) states have the single photon emission signature in their photon statistics and share similar excited state lifetimes. Finally, Figure 3d presents the auto-correlation of the time-course intensity trajectory (Figure 3a) of the single NV center under the excitation of 532 nm and 633 nm lasers. As the single-state photon emission (633 nm excitation) results in a flat curve that is close to zero, the autocorrelation curve of the two-states system (532 nm excitation) clearly suggests a time tag for the correlation. Fitting the autocorrelation curve with Equation (5) will yield the parameter of dark-state probability and the dark-state lifetime. With Equation (6), the transition rate from the “bright” (NV^−^) state to the “dark” (NV^0^) state, i.e., the ionization process, can be extracted as 5.12 ms^−1^, corresponding to a typical time of 0.19 ms. As a comparison, the transition rate from the “dark” (NV^0^) state to the ground state, i.e., the recharging/recombination process, is around 0.05 ms^−1^, and the transition time is 20 ms. The differences between these two transition rates is about 100 times and is usually indistinguishable in the ON/OFF histogram.

### 3.2. The Ionization/Recharging Process of Single NV Centers Is Power-Dependent

To further validate our hypothesis of the “dark” state kinetics, we evaluated the photon emission kinetics of the single NV center when the power (*P*) of the 532 nm excitation is adjusted (Figure 4). At a weak excitation (*P* = 0.02 mW), the intensity trajectory (Figure 4a) showed an intermittency with a large fraction of “dark” states (Figure 4a, photon counting histogram); when the excitation power was increased (*P* = 0.16 mW), the intermittency of the intensity trajectory still existed, but the “bright” states occupied a major fraction. At a high excitation power (*P* = 1.58 mW), the “dark” state vanished, leaving the constant photon emission. Consistently, the NV center exhibited the anti-bunching (single photon) emission statistics, despite the change of the excitation laser power (Figure 4b). However, the fluorescence lifetime of the excited states for the “bright” and “dark” states suggest a differential dependence on the excitation laser power (Figure 4c,d). The lifetime of the “bright” state is independent on the laser power, while the lifetime of the “dark” state increases nonlinearly with the laser power.

The autocorrelation of the intensity trajectories in Figure 4a has yielded a shift of the ACF curves towards the short time with the increased excitation power, as shown in Figure 5a. Fitting these curves with Equation (5) leads to two components of lag time, a slow component in the range of 30–100 ms and a fast component in the range of sub-milliseconds. Given the fact that the slow component has a much smaller fraction (~5%) than the fast component (~95%) towards the total ACF curves and it is independent (both the time and fraction) from the excitation power, we can attribute this slow component to the transition of the NV center to the nanodiamond environment [34]. Therefore, the fast component reflects the transition dynamics between the “bright” and “dark” states exactly, which shows a power-law dependence (slope = −1.61) on the power of excitation (inset, Figure 5a). Interestingly, the transition rates of photon-induced ionization (“bright” state to “dark” state (*k*_12_)) and recharging (“dark” state to ground state (*k*_20_)) yield different responses to the laser power (Figure 5b) [24]. As the excitation laser power increases from 0.02 mW to 1.5 mW, the ionization rate increases linearly from 5.33 ms^−1^ to 160.26 ms^−1^, corresponding to the decrease of the ionization time from 187 μs to 6 μs, which is very similar to the value that was measured by Nuclear Magnetic Resonance (NMR) [16,23]. As a comparison, the recharging rate exhibits a quasi-second-order dependence (index = 1.85) on the laser power. 

Based on the power-dependence of both transition kinetics (*k*_12_ and *k*_20_), it can be concluded that the transition from the “bright” state (NV^−^) to the “dark” states (NV^0^), which is due to the photonionization, is a one-photon process; while the transition from the “dark” states (NV^0^) to the “bright” state (NV^−^) can be considered as a two-photon process, as hypothesized in Figure 1b [15,16]. The observation of the two-photon process for the dark-bright transition was also reported in prior work where sophisticated measurements were conducted [15,16,17,23]. Most importantly, the transition rates measured in our experiments are in excellent agreement with the values reported by others [16,23]. Interestingly, this “two-photon” process allows for a much longer time gap (from tens of microseconds to tens of milliseconds) between the absorption of two consecutive photons, which is significantly different from the regular two-photon process, that requires two simultaneous photon absorptions. This long dwelling time on the “dark” state would offer more flexibility for scientists to control the photon transition of the single NV center.

### 3.3. The Autocorrelation Extracts the Transition Time of Multiple States

As reported by many other works described earlier, the photostability of the single NV center is basically determined by the location of the NV defects in the host-nanodiamond, which is a stochastic process. Therefore, the photon emission of the NV center can be very complicated. Figure 6 illustrates an example in which both 532 nm and 633 nm excitations lead to a multi-state transition. The decomposition of the histogram of intensity trajectories (Figure 6a) suggests that there are approximately three states for the single NV center when being excited by the 633 nm laser; a similar multi-state emission is also observed in the trajectory excited by the 532 nm laser. Although the second-order correlation still suggests the signature of a single photon emission (Figure 6b), the identification of the multi-states can be difficult, and the transition is too fast to distinguish. With the autocorrelation (Figure 6d), we are able to precisely extract the transition time among different states with Equation (5), and finally to obtain the transition kinetics with Equation (6). The calculation suggests that, under the excitation of both 532 nm and 633 nm, there are two transitions from the bright state to the dark states, and the rates are 3.46 μs^−1^ and 27.5 ms^−1^ (532 nm excitation), and 2.65 μs^−1^ and 1 ms^−1^ (633 nm excitation), respectively. The corresponding transition times are 0.29μs and 36.3 μs (532 nm excitation), and 0.38 μs and 1 ms (633 nm excitation). As the longer time can be attributed to the NV^−^ and NV^0^ transition, the short transition time is in the range of sub-microsecond; this can be possibly attributed to the fast radiative recombination of the excited states prior to the photonionization, as similar values were also found in the work reported by Siyushev et al. [15]. 

Comparing with the classical ON-OFF histogram for quantifying the intensity intermittency, the autocorrelation method has various advantages. First of all, a short period (less than 1 s) of intensity trajectory is adequate for the autocorrelation calculation to precisely register the transition time among “bright” and “dark” states. As a comparison, the accuracy of the lifetime of the ON-OFF histogram is highly dependent on the binning size as well as on the duration of the intensity trajectory [27]. Moreover, the lifetime of the ON/OFF states is highly affected by their threshold, while the autocorrelation is independent from the binning size, and no thresholding is required. Furthermore, the autocorrelation illustrates the separation of the short-time transition and the long-time transition in different temporal regions of the autocorrelation curve, enabling the accurate measurement of the small transition time (~10 µs), which is usually vulnerable to noises and a large transition time. Finally, the identification of a multi-state transition is possible with the autocorrelation curve, as it shows distinctive transition times.

## 4. Conclusions

To conclude, this work introduces a simple approach to quantify the kinetics of ON/OFF transition with autocorrelation. Based on a model with triplet states, the analytical expression of the autocorrelation function is derived, and multiple parameters associated with the transition kinetics are extracted. With this new method, we evaluated the photonionization and recharging process of the single NV center in a nanodiamond on the magnesia substrate. While the signature of single photon emission is sustained, the nanodiamond exhibits differential kinetics in response to different excitations (532 nm and 633 nm). Particularly, the nanodiamond has a strong ionization/recharging effect upon excitation of the 532 nm laser, leading to a significant intensity intermittency that indicates “bright” and “dark” states. Our systematic characterization with the autocorrelation reveals that the ionization/recharging effect is power-dependent and is a two-photon process with varying transition times (from tens of microseconds to tens of milliseconds). This work offers a fundamental investigation of the emission property of the nanodiamond, which holds great potential for quantum information science, quantum communication, and quantum bioimaging. 

## 5. Materials and Methods

### 5.1. Preparation of the Nanodiamond

An aqueous suspension of nanodiamonds with a nominal size of 50 nm, 0.5% *w/v*, was obtained from Microdiamant AG (MSY 0.05, GAF, Lengwil, Switzerland). Small nanodiamonds were eliminated, and the suspension was cleaned by centrifuging at 5000 rpm for 5 min followed by supernatant (80% volume fraction) replacement with distilled water. The described procedure was repeated five times. Subsequently, a 20 μL drop of the processed suspension was spin-coated onto a MgO at 2000 rpm for 2 min. Finally, the sample was covered with a 60-nm-thick layer of polyvinyl alcohol (PVA, 1.5% *w/v*) to immobilize and separate the nanodiamonds from oxidization.

### 5.2. Time-Resolved Spectroscopy

A time-resolved single molecule spectroscopy based on a customized scanning confocal microscopy was utilized to record and to characterize the fluorescence emission from the NV centers in the nanodiamonds. Details of the instrument are provided in the Methods section and our previous work [35]. Briefly, a picosecond-pulsed laser beam (LDH-FA, 532 nm, Picoquant Inc. Berlin, Germany) with tunable repetition frequency (10/20 MHz) was delivered onto the sample by a water-immersion objective (NA 1.2, Olympus, Tokyo, Japan) as excitation. Fluorescence emission was collected by the same objective and filtered by a 50 μm pinhole as well as a band pass filter (685-70, Chroma, Vermont US). Then, photons from the NV centers were split via a 50/50 beam-splitter into two identical single photon avalanche photodiodes (SPAD) (SPCM-AQR-14, PerkinElmer Inc. Waltham, Massachusetts, United States). The photon was recorded in the time-correlated single photon counting (TCSPC) module (TimeHarp 200, Picoquant Inc. Berlin, Germany).

The TSCPC histogram synchronized by the laser’s internal clock was fitted by the function $I\left(t\right)=\sum A_{i}\exp\left(-t/\tau_{i}\right)$, to obtain the fluorescence lifetime. Here, *I*(*t*) is the normalized intensity, and *A_i_* (τ_i_) is the amplitude (lifetime) of the *i* th component. The auto-correlation of the recorded fluorescence was performed in the software SynPhoTime (Picoquant Inc. Berlin, Germany). A Hanbury Brown–Twiss (HBT) geometry was used for second-order correlation measurements to investigate the single photon emission property of the NV centers. Details about the setup, the 2nd-order correlation (G2) calculation, and the associated data analysis can be found in [35,36]. 

## Figures and Tables

**Figure 1 nanomaterials-11-00979-f001:**
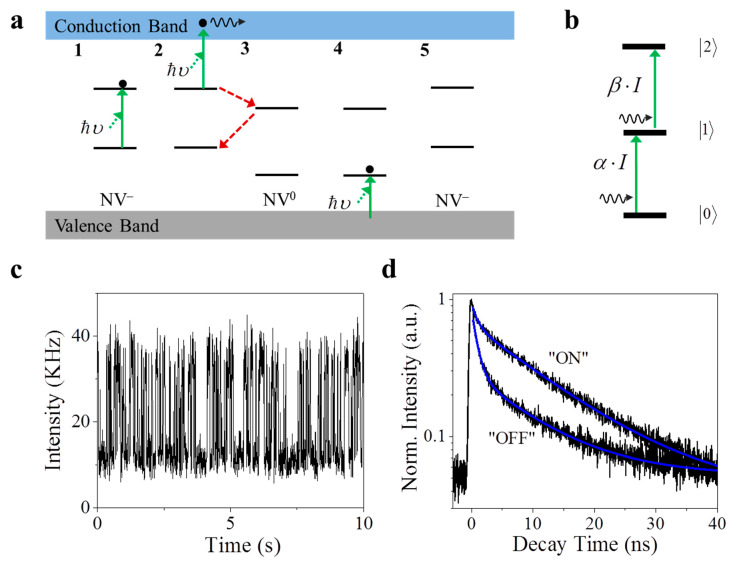
(**a**) Schematic diagram of the charge conversion (photonionization and recharging) process between negative nitrogen vacancy (NV^−^) and neutral nitrogen vacancy (NV^0^). Photonionization process: 1. A single NV^−^ is excited by absorbing one photon with the energy ℏυ. 2. The electron on the excited state of NV^−^ is excited to the conduction band after absorbing a second photon subsequently. 3. The NV center ends up as the NV^0^, which functions as a dark state for NV^−^. Recharging process: 4. An electron in the valence band is excited to the ground state of NV^0^. 5. After obtaining one electron, the NV center ends up as the NV^−^. A similar diagram can be found in [15,16]. (**b**) A diagram summarizing the photonionization is a one-photon absorption process, while recharging/recombination is a sequential two-photon procedure. (**c**) The characteristic time trace of the fluorescence intensity of a single NV center under the excitation of a 532 nm laser beam, showing the fluorescence intermittency in a single NV center. (**d**) A time-correlated single photon counting histogram suggests the lifetime of each state as 11.88 ns for the “ON” state and 6.14 ns for the “OFF” state.

**Figure 2 nanomaterials-11-00979-f002:**
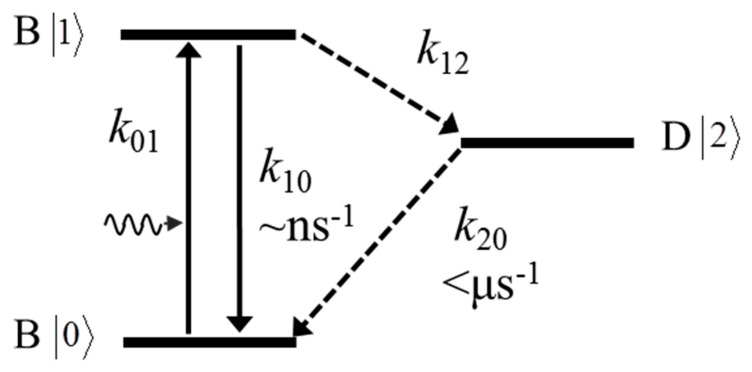
Electronic states of a fluorescence molecule and the transition rate between them. A typical three-level system is presented, including a ground state |0>, an excited state |1>, and a dark state |2>. Refer to the text for a detailed description.

**Figure 3 nanomaterials-11-00979-f003:**
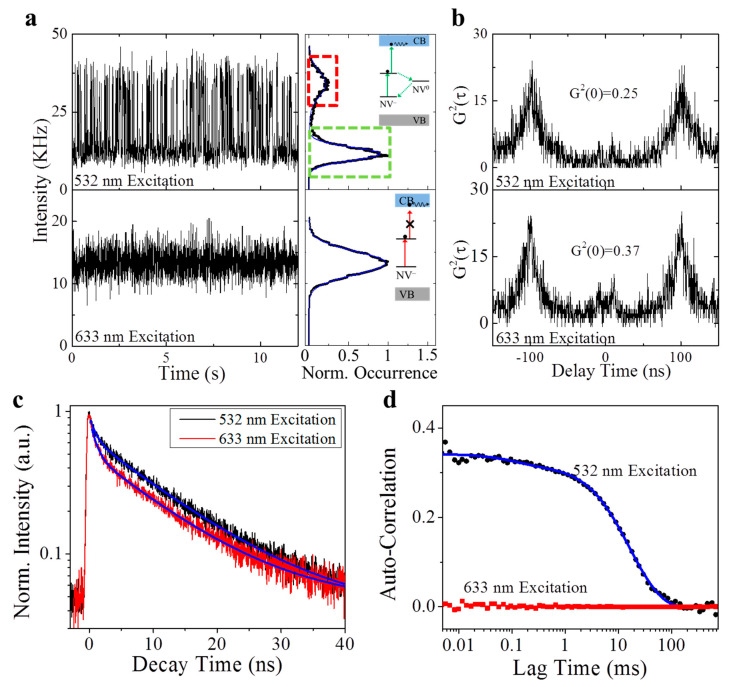
Typical time-resolved spectra of individual NV center under the illumination of 532 nm and 633 nm laser beams. (**a**) A 12-s fragment of a 60 s time trajectory of a single NV center excited by 532 nm laser (upper) and 633 nm laser (lower), with corresponding histograms on the right. The inset is the theoretical explanation; photonionization happens under 532 nm excitation, and stops because the photon energy is smaller than the gap between the excited state and the conduction band. (**b**) The 2nd-order correlation measurement shows the single photon emission signature from the NV center under different excitations. (**c**) Time-correlated single photon counting (TCSPC) decay curves of the fluorescence emission from the NV center under the excitation of 532 nm and 633 nm laser lines suggest a similar lifetime of excited state. (**d**) The auto-correlation calculation of the time trajectory in (**a**) indicates different dark-state dynamics under the illumination of 532 nm and 633 nm laser beams. A two-component mode is required to fit the ACF curve under green illumination, while no dynamics is observed under red illumination.

**Figure 4 nanomaterials-11-00979-f004:**
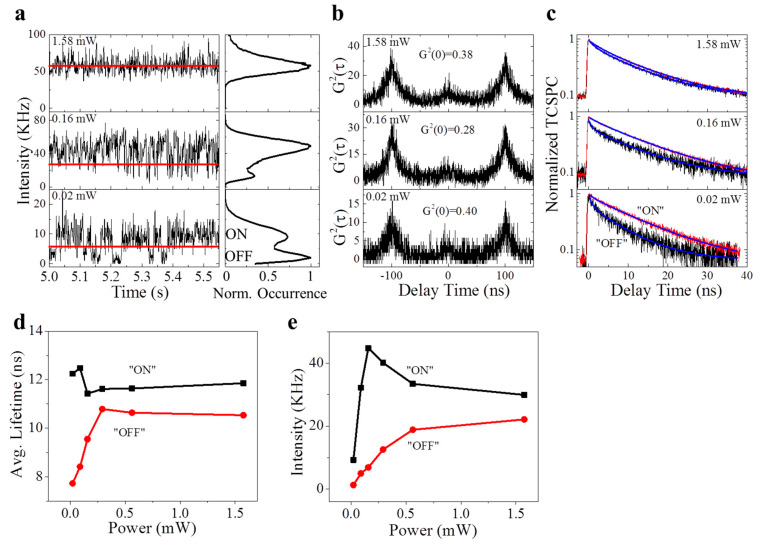
Power-dependent measurement and analysis on the NV center under the illumination of a 532 nm laser beam. (**a**) 0.5-s fragments of a 60-s time trajectory of fluorescence emission from an NV center under the excitation of a 532 nm laser at a different power, and corresponding histograms on the right. The fluorescence intermittency becomes faster at a higher power. Red lines in (**a**) represent the ON/OFF “threshold” for analyzing the emission statistics. (**b**) G2 measurements on the NV center at different laser powers. (**c**) Normalized TCSPC decay curves of “ON” and “OFF” emission states identified in (**a**) at different excitation powers. It becomes difficult to resolve “ON” and “OFF” at a high excitation power. (**d**) Calculated average fluorescence lifetime at different laser powers obtained from (**c**). (**e**) Power-dependent intensity of the “ON” and “OFF” states obtained in (**a**).

**Figure 5 nanomaterials-11-00979-f005:**
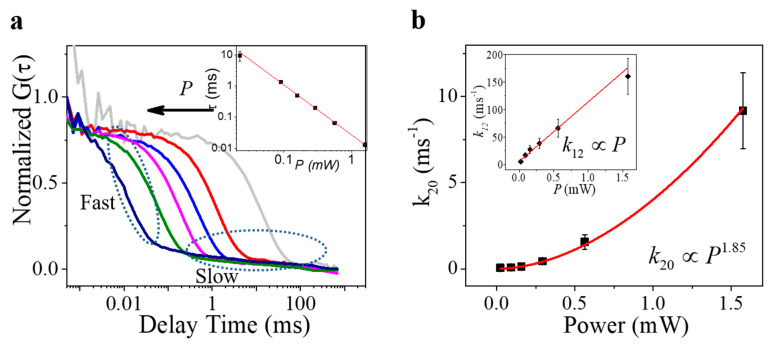
Power-dependent ACF analysis. (**a**) Normalized ACF curves calculated from the time trajectory obtained in Figure 4a. Two components are found in all curves, including a fast component (10~1000 μs) and a slow one (~30 ms). (**b**) Power-dependent |2>-to-|0> transition rate *k*_20_ of the fast component in (**a**), which is fitted by a power equation with the index 1.85; inset plot of the |1>-to-|2> transition rate *k*_12_ versus the excitation laser power.

**Figure 6 nanomaterials-11-00979-f006:**
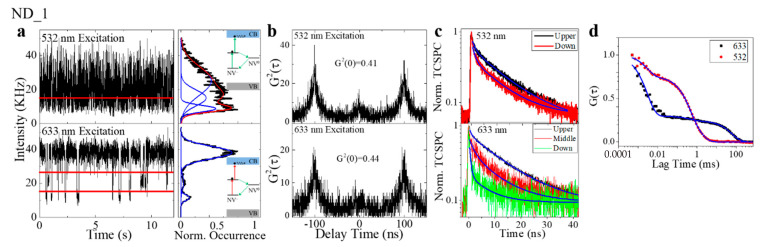
The NV center shows anomalous fluorescence emission phenomena under 532 nm and 633 nm excitations. (**a**) 12-s fragments of a 60-s time trajectory fluorescence emission from a single NV center under the illumination of 532 nm and 633 nm; the histograms are shown correspondingly, and the insets depict the theoretical explanations. Red lines indicate the “threshold” for different states. (**b**) G2 measurements on the NV center under different excitations show the same single photon emission signature. (**c**) Normalized TCSPC decay curves of different states identified in (**a**). (**d**) Normalized ACF curves show different dark-state dynamics under 532 nm and 633 nm excitations.

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
