# Peer review of "Monitoring Dark-State Dynamics of a Single Nitrogen-Vacancy Center in Nanodiamond by Auto-Correlation Spectroscopy: Photonionization and Recharging"

_nanomaterials, 2021, doi:10.3390/nano11040979_

Round 1
Reviewer 1 Report
The authors have studied the photon-induced charge conversion dynamics of a single (NV) center in nanodiamond between two charge states, using the auto-correlation function.
The subject is interesting; however, I have some remarks that should be considered in the revised version:
- The section I introduces the NV centers but I feel it is enough theory and motivations.
The reference: Scientific Reports 10, 1-11 (2020) could be helpful as well the references therein.
- The authors used the classical definition of the autocorrelation function, they should justify why they did not consider the autocorrelation with ordered operators. The reference: Laser Physics Letters 2, 253-257 (2005) and references therein can be helpful.
- I feel that the interpretation of the experimental results is incomplete and needs more details.
Reviewer 2 Report
I find the paper quite interesting and suggest that it can be published after a few corrections.
- Abstract should be completed with the information of the measurements accomplished in these studies.
- Introduction contains Fig.1, which (according to the explanation from the paper) presents the data from the literature. [17, 23, 24]. Hence, the captions under the Figure should be completed with the phrase “with the publisher permit…”, and so on.
- Line 227. The sentence “Our work…” provides nothing new and valuable, so it can be removed from the paper.
- Line 246. Ref.[49] is not on the list of references.
Round 2
Reviewer 1 Report
The revised version is Ok. I accept it.